# Changes of the Protein CoAlation Pattern in Response to Oxidative Stress and Capacitation in Human Spermatozoa

**DOI:** 10.3390/ijms241512526

**Published:** 2023-08-07

**Authors:** Olivia Petrone, Steven Serafini, Bess Yi Kun Yu, Valeriy Filonenko, Ivan Gout, Cristian O’Flaherty

**Affiliations:** 1Department of Pharmacology and Therapeutics, Faculty of Medicine and Health Sciences, McGill University, Montreal, QC H3G 1Y6, Canada; olivia.petrone@mail.mcgill.ca; 2Department of Surgery, Urology Division, Faculty of Medicine and Health Sciences, McGill University, Montreal, QC H3G 1Y6, Canada; steven.serafini@mail.mcgill.ca; 3The Research Institute, McGill University Health Centre, Montreal, QC H4A 3J1, Canada; 4Department of Structural and Molecular Biology, University College London, London WC1E 7JE, UK; bess.yu.15@ucl.ac.uk (B.Y.K.Y.); i.gout@ucl.ac.uk (I.G.); 5Department of Cell Signaling, Institute of Molecular Biology and Genetics, 03680 Kyiv, Ukraine; filonenko@imbg.org.ua; 6Department of Anatomy and Cell Biology, Faculty of Medicine and Health Sciences, McGill University, Montreal, QC H3G 1Y6, Canada

**Keywords:** reactive oxygen species, Coenzyme A, antioxidants, sperm capacitation, spermatozoa, peroxiredoxins, sperm motility, sperm viability

## Abstract

The spermatozoa have limited antioxidant defences, a high polyunsaturated fatty acids content and the impossibility of synthesizing proteins, thus being susceptible to oxidative stress. High levels of reactive oxygen species (ROS) harm human spermatozoa, promoting oxidative damage to sperm lipids, proteins and DNA, leading to infertility. Coenzyme A (CoA) is a key metabolic integrator in all living cells. Recently, CoA was shown to function as a major cellular antioxidant mediated by a covalent modification of surface-exposed cysteines by CoA (protein CoAlation) under oxidative or metabolic stresses. Here, the profile of protein CoAlation was examined in sperm capacitation and in human spermatozoa treated with different oxidizing agents (hydrogen peroxide, (H_2_O_2_), diamide and tert-butyl hydroperoxide (t-BHP). Sperm viability and motility were also investigated. We found that H_2_O_2_ and diamide produced the highest levels of protein CoAlation and the greatest reduction of sperm motility without impairing viability. Protein CoAlation levels are regulated by 2-Cys peroxiredoxins (PRDXs). Capacitated spermatozoa showed lower levels of protein CoAlation than non-capacitation cells. This study is the first to demonstrate that PRDXs regulate protein CoAlation, which is part of the antioxidant response of human spermatozoa and participates in the redox regulation associated with sperm capacitation.

## 1. Introduction

Human infertility affects approximately 1 out of 6 couples worldwide, and 50% of these cases are caused by male factors [1].

Oxidative stress is the net increase in ROS, such as superoxide anion (O_2_^•−^), hydrogen peroxide (H_2_O_2_), nitric oxide (NO^•^) and peroxynitrite (ONOO^−^, a combination of O_2_^•−^ and NO^•^), and in spermatozoa, it can occur either by a decrease in antioxidant defences or by the uncontrolled increased ROS production [2,3]. High ROS levels can cause reduced sperm motility and viability and increase lipid membrane peroxidation and protein and DNA oxidation [4,5,6]. Male infertility is a complex disorder with many causes; a common culprit is the establishment of oxidative stress [4,7]. Astonishingly, ~40% of infertile males have high oxidative stress in their semen [7].

ROS levels that do not decrease sperm viability can impair motility and the ability of the spermatozoon to acquire fertilizing ability through the process of sperm capacitation, suggesting that these vital functions of the spermatozoon are highly sensitive to oxidative stress, generating redox-dependent protein modifications (e.g., S-glutathionylation and tyrosine nitration) of the key proteins involved in these functions in human spermatozoa [6].

Despite elevated ROS levels being harmful to spermatozoa, low ROS amounts are required for sperm capacitation which is defined by a series of biochemical and morphological changes that the spermatozoon must undergo in the oviduct (fallopian tubes in women) to acquire the ability to recognize and fertilize the oocyte [8,9]. These changes include the activation of adenyl cyclase, an increase in intracellular pH and calcium, plasma membrane cholesterol efflux, controlled ROS production, phosphorylation events, etc. [8,9,10]. In addition, tyrosine phosphorylation is a distinctive marker of human sperm capacitation and a late event in the process [11].

Some techniques, such as determining lipid peroxidation, sperm DNA oxidation and redox-dependent protein modifications, have been developed to determine the presence of oxidative stress in the semen of infertile patients. However, these techniques are difficult to implement in the current evaluation of infertile men since they require costly instruments and there are yet to be validated.

PRDXs are antioxidant enzymes that scavenge ROS, such as H_2_O_2_, NO^•^ and ONOO^−^ [12]. PRDXs participate in the redox signalling of human spermatozoa which is necessary to maintain viability and motility and achieve capacitation [5,13]. They are classified into two groups, 2-Cys PRDXs (PRDX1-5) and 1-Cys PRDXs (PRDX6), which differ by the number of cysteine residues in their active site [12]. PRDX6 is the primary antioxidant enzyme in spermatozoa [14]. PRDX6 has peroxidase and calcium-independent phospholipase A_2_ (iPLA_2_) activities, which remove oxidized lipids from the plasma cell membrane [15]. The inhibition of 2-Cys PRDXs or the PRDX6 iPLA_2_ activities increased lipid peroxidation and prevented sperm capacitation in human spermatozoa [13].

It was found that idiopathic infertile patients have lower levels of PRDX1 and PRDX6 in their semen compared to the fertile controls [16]. In human spermatozoa, high ROS levels promote the thiol oxidation of PRDXs, forming disulphide bridges. This protein modification inactivates PRDX activity and promotes other protein modifications such as S-glutathionylation, sulfonation, S-nitrosylation and the formation of 4-HNE adducts [5,6,17].

Coenzyme A (CoA) is a critical molecule for cell metabolism. CoAlation is a novel protein modification characterized by the covalent and reversible addition of CoA to thiol groups on the cysteine residues of proteins under oxidative stress [18]. Bakovíc et al. reported that protein CoAlation of PRDX5 occurs in response to oxidative stress in cardiomyocytes [19]. Moreover, the protein CoAlation of Tau, a microtubule-associated protein in neurons, happens in response to the oxidative stress in HEK293/Pank1β cells relevant for Alzheimer’s disease [20]. However, the role of CoA as an antioxidant in spermatozoa is unknown. Therefore, our objective was to determine the role of protein CoAlation and its regulation by PRDXs in spermatozoa during oxidative stress and sperm capacitation. We hypothesized that protein CoAlation participates in antioxidant protection against oxidative stress, protecting sperm proteins by combining the protein thiol groups with CoASH and redox signalling during capacitation in human spermatozoa.

## 2. Results

### 2.1. Protein CoAlation Pattern Differentially Changed during In Vitro Induced Oxidative Stress in Human Spermatozoa

We observed a dose-dependent increase in the level of protein CoAlation in spermatozoa incubated with H_2_O_2_ for 1 and 3 h (Figure 1A, non-reducing conditions). The increased level of protein CoAlation was abolished when sperm proteins were electrophoresed under reducing conditions, confirming the disulphide bond formation between CoA and cysteine residues and the reversibility of protein CoAlation (Figure 1B). The significant increase in protein CoAlation was only seen at 0.25 mM, 0.5 mM and 1 mM H_2_O_2_ after 1 h in relation to the control (Figure 1A). In addition, a significant increase in protein CoAlation after 3 h was only seen with higher levels of H_2_O_2_ (0.5 mM and 1 mM) in relation to the control (Figure 1B). Interestingly, the induction of protein CoAlation after 1 h of treatment was higher than that obtained after 3 h of incubation, most likely due to the antioxidant response. Sperm proteins of approximately 245 kDa, 180 kDa, 140 kDa, 100 kDa, 75 kDa, 45 kDa and 35 kDa displayed the strongest DTT-dependent CoAlation (Figure 1). Based on these findings and the fact that human spermatozoa are very sensitive to oxidative stress, we can suggest that the increased protein CoAlation might be involved in redox signalling and protecting oxidized surface-exposed cysteine residues from irreversible oxidation.

The treatment of human spermatozoa with diamide and t-BHP also promoted increased levels of protein CoAlation but with different patterns compared to those seen with H_2_O_2_ (Figure 2 and Figure 3). The profile of protein CoAlation in response to diamide was like the one generated with H_2_O_2_, but a dose-dependent induction showed significant differences. The induction of protein CoAlation was seen even at 0.05 mM diamide, which showed a gradual increase with higher concentrations after 1 and 3 h in relation to their control (Figure 2). Similarly to what was observed in H_2_O_2_-treated spermatozoa, the levels of protein CoAlation in diamide-treated spermatozoa seen at 1 h were higher than those observed after 3 h of incubation. Moreover, the diamide-dependent increased protein CoAlation was abolished under reducing conditions.

In contrast to the patterns observed for H_2_O_2_ or diamide treatments, when spermatozoa were incubated with t-BHP, a dose-dependent increase in protein CoAlation was only significantly increased for a 45 kDa protein after 1 h of incubation in non-reducing conditions (Figure 3). However, after 3 h, protein CoAlation levels did not differ in t-BHP-treated spermatozoa compared to their respective controls at the same time of incubation.

Sperm motility was differentially impaired depending on the type of oxidative stress produced. Hydrogen peroxide and diamide produced higher decreases than t-BHP (Figure 4). Although we did see statistical differences between the control and H_2_O_2_ or diamide, none of the compounds decreased sperm viability at levels lower than the 54% value found in the fifth percentile of the semen samples of a population of healthy men (Table 1) [21].

Besides the effect of the different treatments on protein CoAlation and sperm motility over time, we found significant interactions between the treatment with diamide or t-BHP and the time of incubation for protein CoAlation levels (Table 1). Although the *p*-value (0.077) of the interaction between H_2_O_2_ treatment and incubation time was not statistically significant, it may also indicate a trend of the possible effect of treatment and time together on the levels of sperm protein CoAlation. Moreover, sperm viability was affected by treatment with H_2_O_2_ or diamide and the time of incubation, and also by the interaction of treatment and time for H_2_O_2_ and diamide. However, although statistically significant, these effects are not relevant as none of the treatments of incubation time impaired sperm viability since all the samples showed sperm viability higher than the minimum value accepted by the semen analysis guidelines defined by the WHO [21] (Figure 5).

### 2.2. The Inhibition of 2-Cys Peroxiredoxins Increases the Level of Protein CoAlation in Spermatozoa

Since all members of the PRDX family are present and actively respond to high levels of ROS in human spermatozoa, we wanted to study whether the levels of protein CoAlation are affected in spermatozoa challenged with H_2_O_2_ when PRDXs are inhibited. We found that spermatozoa treated with Conoidin A has increased levels of protein CoAlation compared to the controls or to those treated with MJ33 or ethacrynic acid, inhibitors of PRDX6 iPLA_2_ or peroxidase activities, respectively (Figure 6), indicating that CoAlation levels are regulated by 2-Cys PRDXs and that PRDX6 is not critical in the antioxidant response associated with protein CoAlation in human spermatozoa under oxidative stress.

### 2.3. Protein CoAlation Pattern during Time-Course Sperm Capacitation

Lastly, we wanted to know whether the protein CoAlation profile changes during sperm capacitation, the spermatozoon’s process to acquire fertilizing ability. Spermatozoa incubated with FCSu, an in vitro inducer of sperm capacitation, displayed an increase in tyrosine phosphorylation, with a maximal effect seen at 120 min (Figure 7). In addition, when spermatozoa were incubated without FCSu, a higher level of protein CoAlation was observed of a 45 kDa protein in non-capacitating conditions compared to capacitating conditions at any time point (Figure 7). However, only a statistically significant decrease in protein CoAlation was seen in capacitating conditions at 30 min.

## 3. Discussion

In the present study, we demonstrated, for the first time, that human spermatozoa respond to the oxidative stress of various origins by increasing the levels of protein CoAlation, a novel redox-dependent protein modification in response to ROS. Mammalian spermatozoa are very sensitive to oxidative stress mainly due to their limited antioxidant defences [3]. Early studies established that high levels of H_2_O_2_ are detrimental to mammalian spermatozoa, promoting an impairment of energy production and sperm motility [22,23]. Oxidative stress has been identified as one of the culprits of male infertility [4].

Due to high levels of ROS, sperm proteins can be modified by redox-dependent modifications, altering their function [24,25]. We showed that levels of ROS that do not impair viability significantly reduced the capacitation and motility of human spermatozoa and that high levels of S-glutathionylation and tyrosine nitration were associated with this impairment [6,26]. Here, we showed that different sources of in vitro oxidative stress, capable of impairing sperm motility (without modifying sperm viability), promoted increased levels of protein CoAlation, suggesting that this novel redox-dependent protein modification is part of the antioxidant response in human spermatozoa. The fact that the protein CoAlation levels at 3 h are lower than those observed at 1 h provides evidence of the reversibility of this modification. It is possible that antioxidant enzymes scavenge ROS, lowering their levels and reducing those proteins that were protected by CoAlation to be active in sperm functions again. However, these experiments were carried out under in vitro conditions, and further studies using a mouse model must be conducted to confirm that this reversibility also occurs in vivo.

Diamide is a superoxide anion (O_2_^•−^) donor, a free radical that is converted to H_2_O_2_ by either superoxide dismutase (SOD) or spontaneously. Human spermatozoa contain limited amounts of cytoplasmic and negligible quantities of mitochondrial SOD to scavenge O_2_^•−^ and generate more H_2_O_2_ [27]. Hydrogen peroxide causes a higher increase in protein CoAlation and subsequent oxidative stress than diamide (Figure 1 and Figure 2). Therefore, O_2_^•−^ is most likely reacting with biomolecules (perhaps nitric oxide to form ONOO^−^), causing less O_2_^•−^ to be converted to H_2_O_2_, resulting in a lower net amount of H_2_O_2_ in diamide-induced oxidative stress. Notably, the increase in protein CoAlation in non-reducing conditions was higher at the 1-h incubation than the 3-h incubation. This could be due to antioxidant enzymes being reactivated to scavenge ROS, decreasing the oxidative stress present at the 3-h incubation. Samples were incubated for 3 h because the duration of capacitation is 3.5 h, where the 1-h time point acts as an early time point during capacitation [8,11].

Importantly, protein CoAlation levels in spermatozoa remained constant in reducing conditions in the presence of H_2_O_2_-_,_ diamide- or t-BHP-induced oxidative stress. These results confirm the reversible nature of the CoAlation modification of thiol groups present in sperm proteins. The fact that protein CoAlation increases dose-dependently to oxidative stress-generating compounds indicates that it is a protective mechanism similar to what we observed previously with S-glutathionylation, another reversible protein modification in response to oxidative stress generated by the H_2_O_2_ treatment of human spermatozoa [6].

PRDXs, the main antioxidant enzymes of spermatozoa, have one or two thiol groups in their active site [28]; thus, they are susceptible to redox-dependent modifications in spermatozoa are under oxidative stress. When thiol groups of PRDXs are oxidized, the enzymes become inactive [28]. The reversible nature of some of these oxidations, such as those observed in PRDX1 to 5, can be overcome by the reduction activity of the thioredoxin/thioredoxin reductase (TRD)/NADPH system. Indeed, when this system is inhibited by auranofin, an inhibitor of TRD, or when the NADPH supply is prevented by the inhibition of NADP-isocitrate dehydrogenase, malic enzyme or glucose 6-phosphate dehydrogenase, a decrease in sperm viability along with increased levels of mitochondrial O_2_^•−^ production and the reduction of mitochondrial membrane potential were observed. These findings indicate the need for 2-Cys PRDXs to fight against oxidative stress in spermatozoa [5]. As it was previously reported that PRDX5 CoAlation occurs in cardiomyocytes under oxidative stress, protein CoAlation may occur in spermatozoa to protect PRDXs during oxidative stress [19]. In addition, antioxidant enzymes in spermatozoa, such as PRDXs, have been shown to endure protein modifications during oxidative stress, such as thiol oxidation, as we described previously [17].

During oxidation, PRDXs are capable of aggregating to form high molecular mass complexes in healthy human spermatozoa treated with high concentrations of H_2_O_2_ [17] or spermatozoa from infertile patients [16] but not with t-BHP [29]. These observations indicate the different responses observed in spermatozoa depending on the prevalent ROS in the incubation medium. We observed that H_2_O_2_ or diamide-induced oxidative stress produces a protein CoAlation of several prominent bands of approximately 75 kDa, 140 kDa and 245 kDa in non-reducing conditions (Figure 1 and Figure 2). The PRDX antioxidant enzyme family exhibits a molecular size between 20 to 31 kDa [12], whereas CoA is 767 Da [18]. Since PRDXs have been reported to form high molecular mass complexes in response to oxidative stress [17], PRDXs could be forming complexes through the disulphide bridges with CoA, accounting for the large protein bands seen in non-reducing conditions observed here (Figure 1 and Figure 2).

Human spermatozoa are better equipped with antioxidants to respond against organic hydroperoxides compared to H_2_O_2_. Human spermatozoa contain high amounts of PRDX6, which is the primary antioxidant defence in these cells [14]. PRDX6 is a unique member of the PRDX family as it is the only one that has both peroxidase and iPLA_2_ activity [15]. Through these two activities, PRDX6 protects mouse and human spermatozoa against oxidative stress [5,13,30,31]; we demonstrate this critical role of PRDX6 by the pharmacological inhibition of these activities or using the PRDX6 knockout mouse model [32]. There is no compensatory mechanism when one of these activities is not present, as we found that mice lacking either peroxidase activity (C47s knock-in mouse strain) or iPLA_2_ activity (knock-in D140A strain) are subfertile, with spermatozoa displaying significant oxidative damage characterized by high levels of lipid peroxidation, 8-deoxyguanosine and protein tyrosine nitration [33].

Since human spermatozoa contain all six members of PRDXs, we wanted to address the question of whether the inhibition of 2-Cys PRDXs or PRDX6 modifies the protein CoAlation pattern we observed when spermatozoa were exposed to oxidative stress (Figure 6). Conoidin A inhibits 2-Cys PRDXs, whereas MJ33 and ethacrynic acid inhibit the PRDX6 iPLA_2_ and the re-activation of PRDX6 peroxidase activity, respectively [5]. We used two concentrations of H_2_O_2_, 0.05 mM H_2_O_2_, that do not impair sperm viability nor motility and promotes sperm capacitation, and 0.1 mM H_2_O_2_, which induces oxidative stress without damaging sperm viability but significantly reduces sperm motility and capacitation [6,34]. Interestingly, we found that protein CoAlation significantly increased only when Conoidin A was present in the incubation medium, suggesting that 2-Cys PRDXs regulate the protein CoAlation levels in human spermatozoa. Astonishingly, protein CoAlation does not increase when the PRDX6 iPLA_2_ activity is inhibited, the reactivation of PRDX6 peroxidase activity is inhibited or PRDX6 activity is entirely inhibited, suggesting that 2-Cys PRDXs are enough to regulate the levels of protein CoAlation in spermatozoa (Figure 6). PRDX6 is, therefore, not essential in preventing protein CoAlation and subsequent oxidative stress, which is surprising since PRDX6 is the most critical antioxidant enzyme in spermatozoa to protect sperm DNA and avoid lipid peroxidation and tyrosine nitration [14]. Altogether, these findings indicate that 2-Cys PRDXs are essential in regulating protein CoAlation in spermatozoa under oxidative stress and provide more understanding of the antioxidant system rule by the PRDX family in human spermatozoa. Further studies are required to understand the role of protein CoAlation at the molecular level as part of the antioxidant response in human spermatozoa.

Finally, we were interested in studying the participation of protein CoAlation during sperm capacitation, a process essential for fertilization involving mild oxidative stress and promoting low lipid peroxidation levels [13,35]. When spermatozoa were incubated under non-capacitating conditions (without FCSu), higher levels of protein CoAlation were observed in a single protein band of ~45 kDa compared to those observed in spermatozoa under capacitating conditions at any time point (Figure 7). We found that this protein CoAlation was decreased in capacitated sperm with respect to non-capacitated sperm; however, only a significant decrease in protein CoAlation was observed at 30 min in capacitating conditions.

It is known that low and specific ROS amounts are required for sperm capacitation to occur [10,36]. ROS production is a tightly regulated early event of capacitation that causes the activation of protein kinases, ultimately inducing important phosphorylation events [37]. For example, ROS activate adenyl cyclase, which activates protein kinase A (PKA) [38] through increasing cAMP levels, which causes, among other players, the essential tyrosine phosphorylation event of capacitation [37]. Thus, increased protein CoAlation at 45 kDa in non-capacitating conditions suggests that this modification protects proteins with that molecular weight and that during capacitation, they are de-CoAlated and, therefore, susceptible to be oxidized by low levels of ROS produced at the beginning of capacitation. It is striking that solely a 45 kDa-band of similar molecular weight to that of the regulatory subunit of PKA found in spermatozoa [39] is seen in Figure 7, and that a significant decrease in its CoAlation levels was observed at the beginning of capacitation. Based on these findings, we can hypothesize that PKA activity is regulated by protein CoAlation since the maximal activity in human spermatozoa was found at 30 min after the beginning of capacitation [38]. Therefore, we can speculate that protein CoAlation plays a role in redox signalling to ensure human sperm capacitation. However, further studies are necessary to identify the 45 kDa protein (by mass spectrometry analysis) and determine whether PKA activity is regulated by protein CoAlation during capacitation.

In conclusion, protein CoAlation participates in the antioxidant protection of human spermatozoa during oxidative stress, and 2-Cys PRDXs play a role in regulating this redox-dependent protein modification in human spermatozoa. Moreover, protein CoAlation modulation is needed to ensure capacitation, thus suggesting the participation of this protein modification in redox signalling necessary for the spermatozoon to acquire fertilizing capabilities. This study opens a new avenue for further studies to decipher the role of protein CoAlation in the antioxidant response and redox signalling necessary for male fertility.

## 4. Materials and Methods

### 4.1. Materials

Percoll was purchased from GE Healthcare (Montreal, QC, Canada). The nitrocellulose membranes (pore size, 0.2 μm) were purchased from Cytiva (Marlborough, MA, USA). PierceTM ECL Western Blotting Substrate was purchased from ThermoScientific (Rockford, IL, USA). Foetal Cord Serum Ultrafiltrate, FCSu, was prepared from foetal cord serum obtained from the fetal cord blood bank at the Royal Victoria Hospital (Montreal, QC, Canada). The mouse monoclonal anti-CoA primary antibody was developed by Dr. Ivan Gout, University College London, UCL (London, UK). The mouse monoclonal anti-α-tubulin and mouse monoclonal anti-phospho-tyrosine (clone 4G10) antibodies were purchased from Sigma Aldrich (Oakville, ON, Canada). The peroxidase-conjugated goat anti-mouse secondary antibody was purchased from Jackson ImmunoResearch Laboratories Inc. (Bar Harbor, ME, USA). In addition, all reagents and inhibitors were purchased from Sigma-Aldrich (Oakville, ON, Canada).

### 4.2. Sperm Sample Preparation

Semen samples were provided by healthy fertile volunteers between the ages of 18 and 30 after 72 h of sexual abstinence. Healthy donors were recruited at the McGill University Health Centre. After semen collection, the samples were incubated at 37 °C for 30 min to allow their liquefaction. Samples were then placed in a four-layer Percoll gradient of differing concentrations, 95%, 65%, 40%, to 20%, and centrifuged at 2300× *g* for 30 min at 20 °C. After centrifugation, spermatozoa were taken from the 95% layer and the 95–65% interphase to obtain the healthiest and most highly motile spermatozoa. Sperm concentration was determined using a hemocytometer chamber. Sperm samples were then diluted to 250 × 106 sperm/mL in a Biggers, Whitten and Whittingham medium (BWW, pH 7.8) [40] and subsequently used in the following experiments.

### 4.3. Induction of In Vitro Oxidative Stress by Hydrogen Peroxide, Diamide and Tert-Butyl Hydroperoxide

Highly motile Percoll-selected human spermatozoa at a concentration of 50 × 106 sperm/mL were incubated in BWW medium with or without different concentrations, 0, 0.05, 0.1, 0.25 and 1 mM H_2_O_2_, diamide (O_2_^•−^ donor) or tert-butyl hydroperoxide (t-BHP) for 1 and 3 h at 37 °C. After the incubation, the samples were split into two aliquots and sperm proteins were supplemented in electrophoresis sample buffer with or without 100 mM dithiothreitol (DTT, reducing and non-reducing conditions, respectively), boiled and centrifuged to recover the supernatant to determine protein CoAlation.

### 4.4. Inhibition of PRDXs in Spermatozoa Treated with H_2_O_2_ to Promote In Vitro Oxidative Stress 

Highly motile Percoll-selected spermatozoa at a concentration of 50 × 106 sperm/mL were incubated in BWW medium at 37 °C for 15 min with or without the following PRDX inhibitors: 0.04 mM Conoidin A, 0.02 mM MJ33 and 0.4 mM Ethacrynic acid, inhibitors of 2-Cys PRDXs, PRDX6 iPLA_2_ activity and a reduced glutathione adduct, respectively [5]. After the 15-min incubation to allow the inhibitors to enter spermatozoa, 0.05 or 0.1 mM H_2_O_2_ was added to the samples and incubated at 37 °C for 1 h. These concentrations do not impair viability. We used these concentrations to test the participation of PRDXs during the induction of sperm capacitation with 0.05 mM H_2_O_2_ or during an oxidative stress generated by 0.1 mM H_2_O_2_ that impairs sperm motility. After the incubation, samples were prepared for CoAlation determination by immunoblotting.

### 4.5. Sperm Viability and Motility Analysis

After spermatozoa were subjected to in vitro H_2_O_2_-, diamide- or t-BHP -induced oxidative stress for 1 and 3 h at 37 °C, we assessed sperm viability and motility using the hypoosmotic swelling (HOST) and a computer-aided sperm analysis system CASA), respectively, as done before [6]. After treatments, a sperm aliquot of each sample was used to determine total and progressive motilities at 37 °C by analyzing more than 200 cells per sample using the CASA system IVOS II (Hamilton Thorne, Beverly, MA, USA), equipped with a 10× negative phase contrast objective and temperature control. The videos were taken at a frame rate of 60 Hz frames: 1–60. Another sperm aliquot was centrifuged and resuspended in a hypo-osmotic buffer containing 1.5 mM fructose and 1.5 mM sodium citrate (pH 7.4), incubated for 30 min at 37 °C, and placed onto Superfrost slides (Fischer Scientific, Ottawa, ON, Canada). Then, at least 200 cells were analyzed using a Leica DFC 450C microscope at 200× magnification with Leica Application Suite X (LAS X) software version 1.1.0.12420 (Leica Microsystems, Wetzlar, Germany) to determine the percentage of viable sperm with or without different degrees of curly tails or the presence of a droplet [41,42].

### 4.6. Profile of Protein CoAlation during Time-Course Sperm Capacitation

Highly motile Percoll-selected spermatozoa at a concentration of 50 × 106 sperm/mL were incubated in BWW medium at 37 °C, with or without FCSu, a well-known in vitro capacitation inducer [43], to obtain capacitating and non-capacitating conditions, respectively. As done previously, the samples were incubated at different intervals of 0, 5, 15, 30, 60 and 120 min [44]. After each incubation, the samples were split into non-reducing and reducing conditions to determine levels of protein CoAlation and tyrosine phosphorylation, respectively, by immunoblotting.

### 4.7. SDS PAGE and Immunoblotting

After the preceding treatments, each sperm sample was divided into two aliquots to be supplemented with electrophoresis buffer containing 100 μM vanadate, 20 mM β-glycerolphosphate, 5 mM sodium fluoride with or without dithiothreitol (DTT; reducing and non-reducing conditions, respectively). Sperm samples were supplemented with electrophoresis buffer and 100 mM DTT (reducing conditions) or treated with 50 mM N-ethylmaleimide (NEM) 10 min on ice to avoid further oxidation of thiols groups in sperm proteins, and then supplemented with sample buffer without DTT (non-reducing conditions). All samples were then vortexed, boiled at 100 °C for 5 min, and centrifuged at 21,000× *g* for 5 min at 20 °C.

The samples were then electrophoresed in a 10% SDS–polyacrylamide gel, electrotransferred onto a nitrocellulose membrane and then the membrane was blocked in 5% skim milk in 2 mM Tris (pH 7.8)-buffered saline with 0.1% Tween 20 (TTBS) for 1 h. The membrane was incubated with the monoclonal mouse anti-CoA antibody (1:3000) overnight at 4 °C. The incubations with monoclonal mouse anti-tyrosine phosphorylation primary antibody (1:10,000) and mouse α-tubulin antibody (1:10,000) were conducted at room temperature for 1 h. Then, the membranes were washed with TTBS, five times for 8 min. Finally, the membrane was then incubated with the secondary antibody (1:2500) for 45 min at 20 °C and then washed using TTBS five times for 8 min. The membranes were lastly incubated in Enhanced Chemiluminescence (ECL) substrate for 5 min and immediately imaged using an Amersham Imager600 (General Electric Healthcare Bio-Sciences Corp., Piscatway, NJ, USA) on the Chemiluminescence setting to detect positive immunoreactive bands.

The relative intensities of protein bands per sample were assessed using Fiji Image J version 1.46r (National Institutes of Health, Bethesda, MD, USA) and standardized to that of α-tubulin (the loading control) by dividing the relative intensity of the protein band of interest. The relative intensities were then normalized to the respective control, and subsequently, the average relative intensity and standard error were determined for each experiment.

### 4.8. Statistical Analysis

All data are presented as the mean ± SEM. Shapiro–Wilk and Levene’s tests assessed normal data distribution and variance homogeneity, respectively. The statistical differences in protein CoAlation and tyrosine phosphorylation levels among samples and the effects of treatment or incubation times were evaluated by two-way ANOVA and Tukey test (*p* ≤ 0.05).

## Figures and Tables

**Figure 1 ijms-24-12526-f001:**
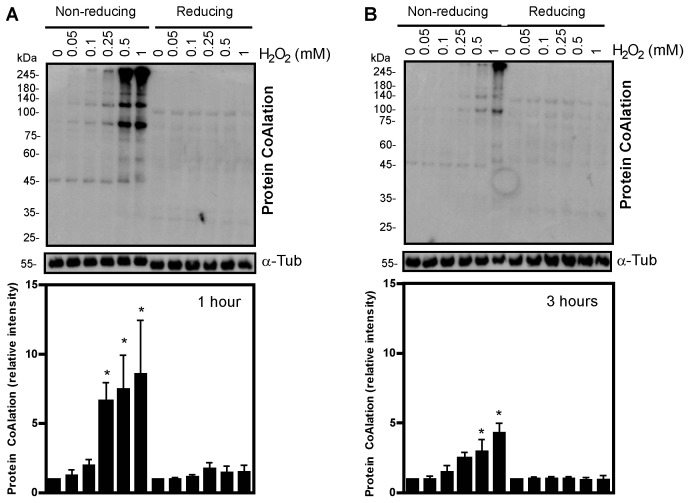
Protein CoAlation pattern in spermatozoa treated with increasing concentrations of H_2_O_2_ under non-reducing conditions. Dose-dependent increased levels of protein CoAlation under non-reducing conditions and loss of protein CoAlation signal in reducing conditions after (**A**) 1 h and (**B**) 3 h of incubation. Spermatozoa were incubated in BWW medium at 37 °C with different H_2_O_2_ concentrations, and tubes were split into non-reducing and reducing conditions. The samples were electrophoresed in SDS–polyacrylamide gel, electrotransferred onto a nitrocellulose membrane and immunoblotted with a monoclonal mouse anti-CoA antibody. The average relative intensities per sample were measured using Fiji Image J and standardized to the loading control, α-tubulin, under reducing conditions. The results are presented as mean ± SEM and represent sperm samples from different healthy donors (n = 4). Two-way ANOVA and Tukey test, * different from 0 mM, *p* ≤ 0.05.

**Figure 2 ijms-24-12526-f002:**
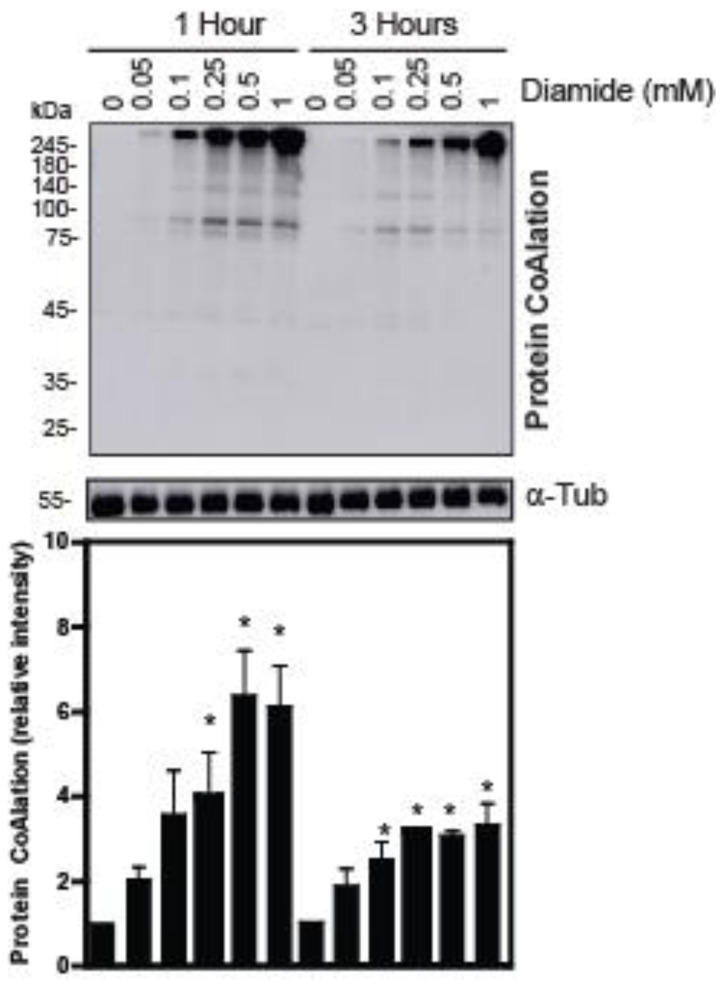
Protein CoAlation pattern in spermatozoa treated with increasing concentrations of diamide. Protein CoAlation levels dose-dependently increased, as seen under non-reducing conditions at both times of incubations. Spermatozoa were incubated in BWW medium at 37 °C, and tubes were split into non-reducing and reducing conditions. The samples were electrophoresed in SDS–polyacrylamide gel, electrotransferred onto a nitrocellulose membrane and immunoblotted with a monoclonal mouse anti-CoA antibody. The average relative intensities per sample were measured using Fiji Image J and standardized to the loading control, α-tubulin, under reducing conditions. The results are presented as mean ± SEM and represent sperm samples from different healthy donors (n = 4). Two-way ANOVA and Tukey test, * different from 0 mM, *p* ≤ 0.05.

**Figure 3 ijms-24-12526-f003:**
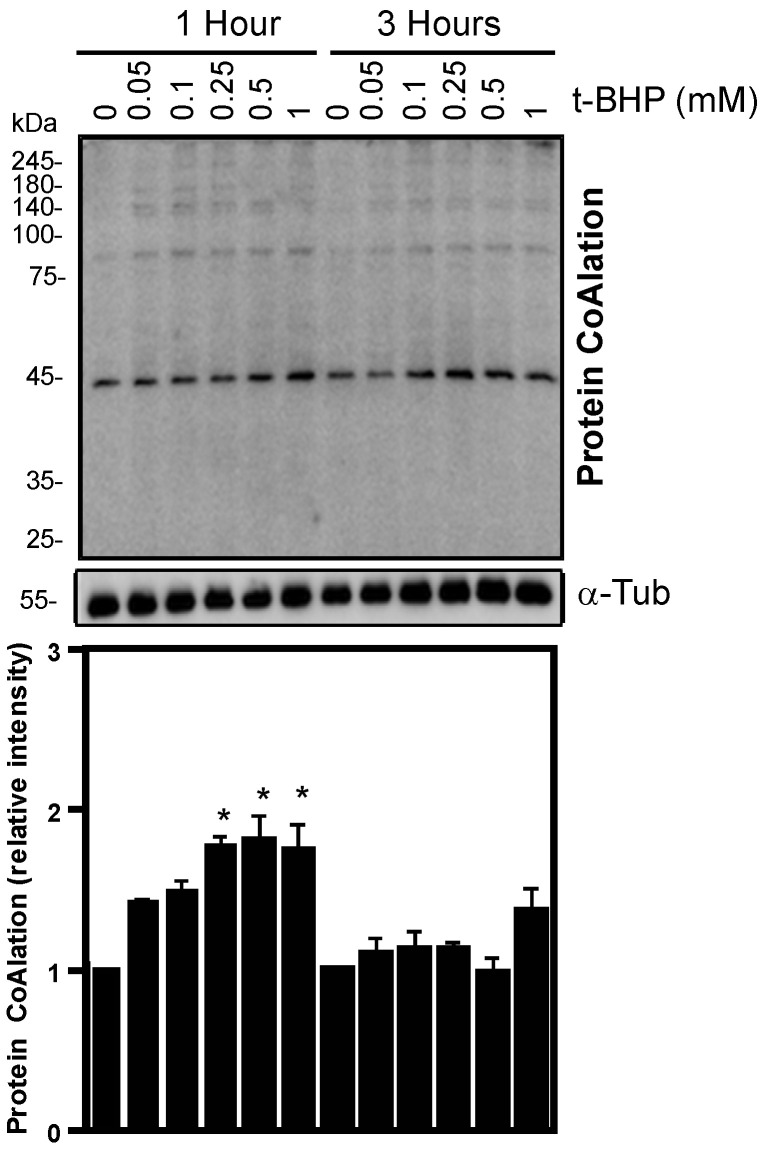
Protein CoAlation pattern in spermatozoa treated with increasing concentrations of tert-butyl hydroperoxide (t-BHP). Dose-dependently increased levels of protein CoAlation at 45 kDa after 1 h (2) decreased levels of protein CoAlation at 45 kDa with high concentrations of t-BHP hydroperoxide after 3 h in samples under non-reducing conditions. Spermatozoa were incubated in BWW medium at 37 °C for 1 h and 3 h. Then, the samples were electrophoresed in SDS–polyacrylamide gel, electrotransferred onto a nitrocellulose membrane and immunoblotted with a monoclonal mouse anti-CoA antibody. The average relative intensities per sample were measured using Fiji Image J and standardized to the loading control, α-tubulin, under reducing conditions. The results are presented as mean ± SEM and represent sperm samples from different healthy donors (n = 4). Two-way ANOVA and Tukey test, * different from 0 mM, *p* ≤ 0.05.

**Figure 4 ijms-24-12526-f004:**
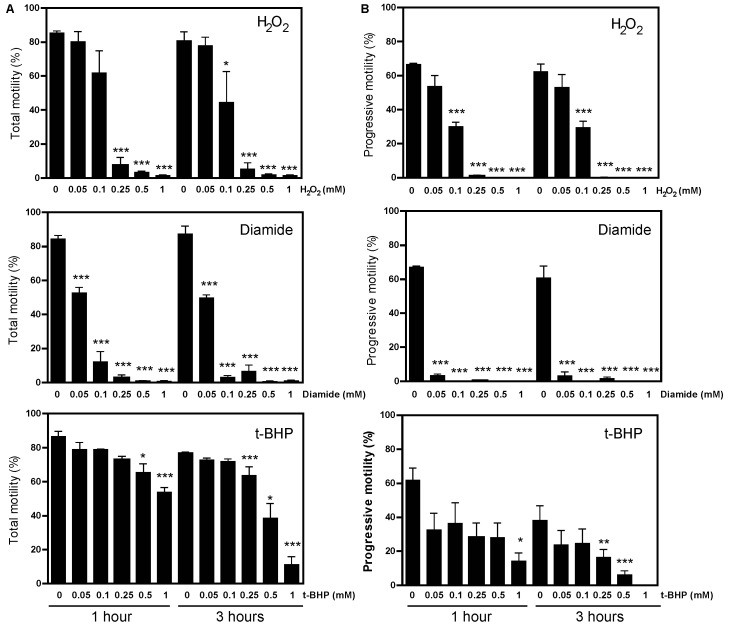
Effects of H_2_O_2_, diamide and t-BHP treatments on sperm motility. Spermatozoa were incubated with different concentrations of H_2_O_2_, t-BHP or diamide and total (**A**) and progressive (**B**) motility was assessed at 1 and 3 h at 37 °C. The results are presented as mean ± SEM and represent sperm samples from different healthy donors (n = 4). Two-way ANOVA and Tukey test, *, ** and *** mean significantly different from 0 mM sample (*p* ≤ 0.05, *p* ≤ 0.01 and *p* ≤ 0.001, respectively).

**Figure 5 ijms-24-12526-f005:**
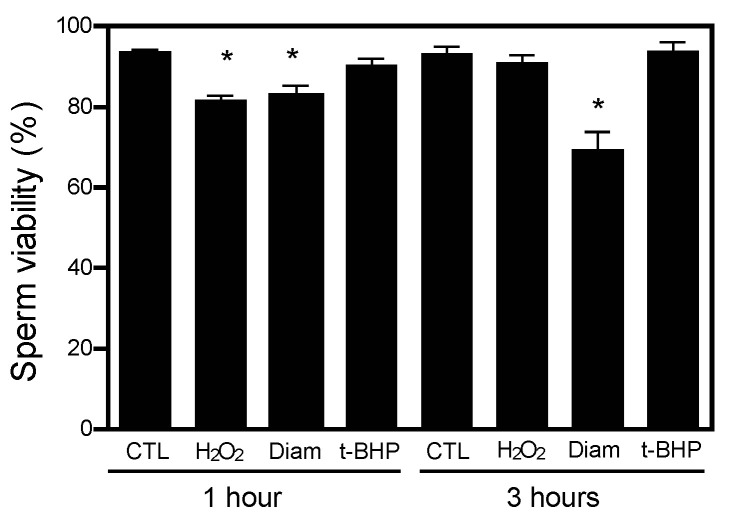
Sperm viability in spermatozoa control and treated with the oxidative stress generators incubated for 1 and 3 h at 37 °C. Human sperm viability was measured in spermatozoa controls (CTL) or treated with 1 mM H_2_O_2_, diamide (Diam) or t-BHP at 1 and 3 h of incubation. The results are presented as mean ± SEM and represent sperm samples from different healthy donors (n = 4). Two-way ANOVA and Tukey test, * different from 0 mM at 1 h of incubation, *p* ≤ 0.05. Note that the sperm viability values in all samples were higher than 54%, the minimum value accepted in normal human semen according to the World Health Organization [21].

**Figure 6 ijms-24-12526-f006:**
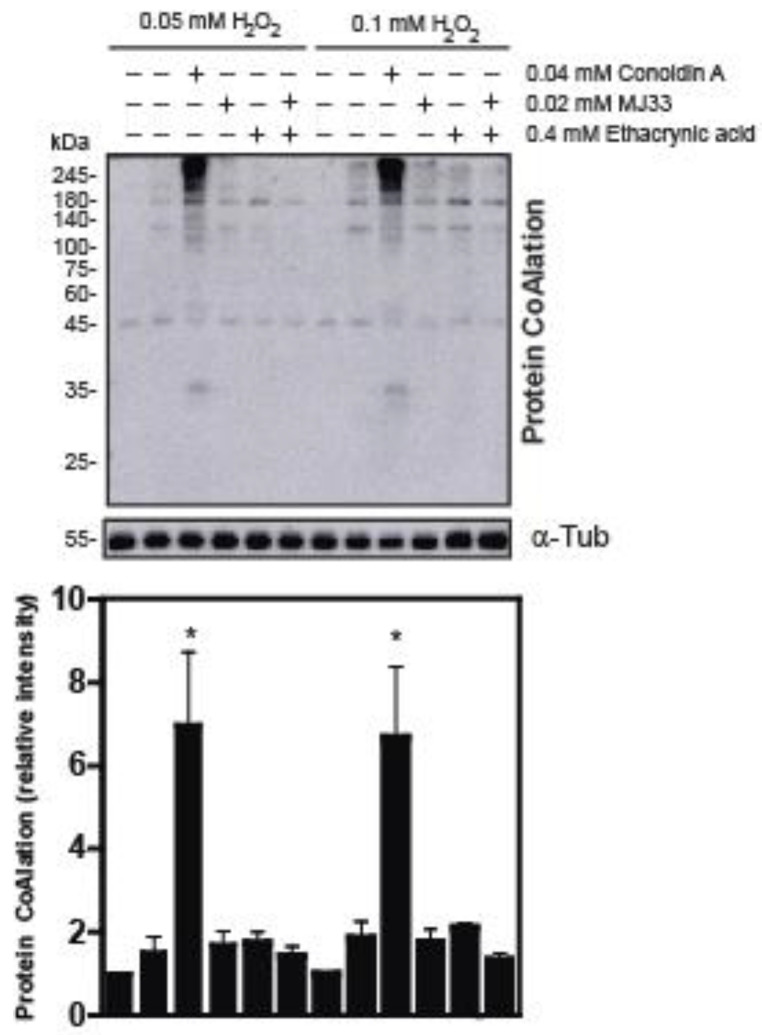
2-Cys peroxiredoxins regulate protein CoAlation in human spermatozoa. Spermatozoa treated with or without PRDX inhibitors in the presence of 50 or 100 μm H_2_O_2_ show increased protein CoAlation in the presence of Conoidin A, a 2-Cys PRDX inhibitor, but when PRDX6 peroxidase or iPLA_2_ activities were inhibited with ethacrynic acid or MJ33, respectively, in sperm proteins under non-reducing conditions. Spermatozoa were incubated in BWW medium at 37 °C for 1 h, and tubes were made into non-reducing conditions. The samples were electrophoresed in SDS–polyacrylamide gel, electrotransferred onto a nitrocellulose membrane and immunoblotted with a monoclonal mouse anti-CoA antibody. The average relative intensities per sample were measured using Fiji Image J and standardized to the loading control, α-tubulin, under reducing conditions. The results are representative of sperm samples from five different healthy donors (n = 4). Two-way ANOVA and Tukey test, * higher than all others *p* ≤ 0.05.

**Figure 7 ijms-24-12526-f007:**
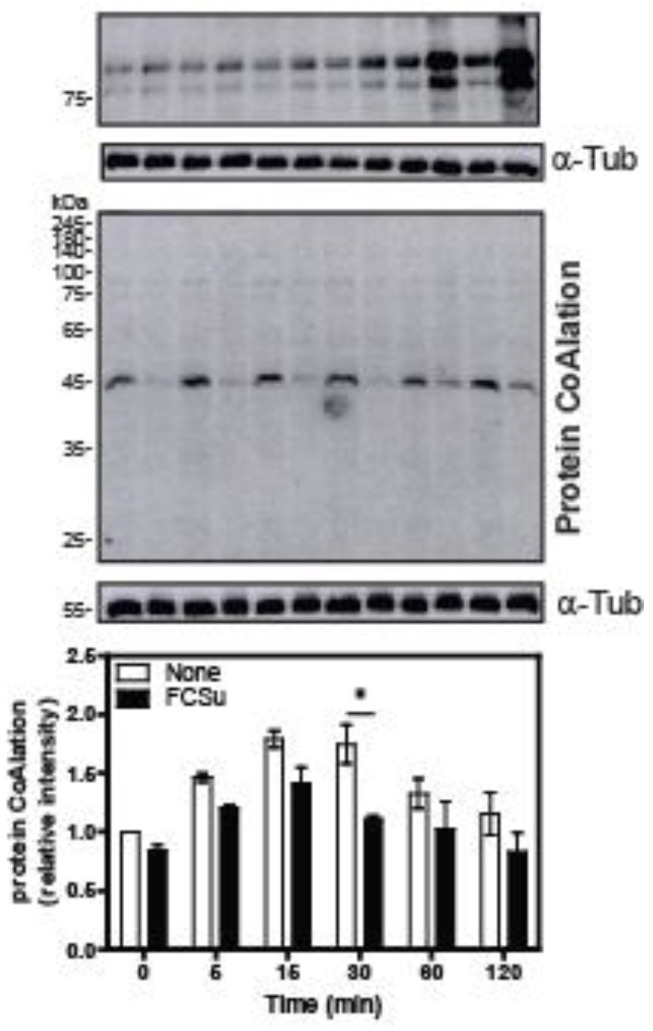
Protein CoAlation pattern during human sperm capacitation. Spermatozoa treated with or without FCSu being capacitating (+) and non-capacitating (−) conditions, respectively, show decreased levels of protein CoAlation in capacitating conditions at 30 min and increased levels of tyrosine phosphorylation maximally at 2 h under capacitating conditions. Spermatozoa were incubated in BWW medium at 37 °C for different time points of up to 2 h, and tubes were split into non-reducing and reducing conditions. The samples were electrophoresed in SDS–polyacrylamide gel and electrotransferred onto a nitrocellulose membrane. The non-reducing membrane was immunoblotted with a monoclonal mouse anti-CoA antibody, whereas the reducing membrane was immunoblotted with a monoclonal mouse anti-tyrosine-phosphorylation antibody. The average relative intensities per sample were measured using Fiji Image J and standardized to the loading control, α-tubulin. The results are representative of sperm samples from three different healthy donors (n = 3). Two-way ANOVA, and Tukey test, * statistically different, *p* ≤ 0.05.

**Table 1 ijms-24-12526-t001:** Complete summary of statistical analysis by two-way ANOVA for each treatment.

Source of Variation	Protein CoAlation Levels	Sperm Viability	Total Motility	Progressive Motility
H_2_O_2_ treatment	*p* < 0.001	*p* = 0.002	*p* < 0.0001	*p* < 0.0001
time	*p* = 0.0007	*p* = 0.03	*p* = 0.20	*p* = 0.601
H_2_O_2_ treatment × time	*p* = 0.077	*p* = 0.014	*p* = 0.83	*p* = 0.99
Diamide treatment	*p* < 0.0001	*p* = 0.003	*p* < 0.0001	*p* < 0.0001
time	*p* < 0.0001	*p* = 0.04	*p* = 0.60	*p* = 0.47
Diamide treatment × time	*p* = 0.0151	*p* = 0.06	*p* = 0.45	*p* = 0.57
t-BHP treatment	*p* < 0.0001	*p* = 0.49	*p* < 0.0001	*p* < 0.0001
time	*p* < 0.0001	*p* = 0.45	*p* = 0.61	*p* < 0.001
t-BHP treatment × time	*p* = 0.0006	*p* = 0.29	*p* = 0.24	*p* = 0.89

## Data Availability

The data that support the findings of this study are available from the corresponding author upon reasonable request.

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
