# Peer review of "Changes of the Protein CoAlation Pattern in Response to Oxidative Stress and Capacitation in Human Spermatozoa"

_ijms, 2023, doi:10.3390/ijms241512526_

Round 1

Reviewer 1 Report

This study focuses on the role of CoAlation and its regulation via PRDX in the human spermatozoon during capacitation and in the context of oxidative stress.

The introduction describes the role of oxidative stress in pathological situations. The mechanisms are explained. However, the role of oxidative stress in the capacitation process needs to be better explained. In addition, the authors point out that fertility clinics do not measure ROS levels and that this could explain infertility. This statement is not correct. Several laboratories have developed tests to measure ROS levels using different methods (e.g. assay of oxidative stress markers (e.g. MDA) or marking by immunohistochemistry). These methods should be described and the results commented on.

The results clearly show the protein CoAlation pattern in electrophoresis under reducing or non-reducing conditions. Spermatozoa were stressed by exposure to H2O2. However, the sperm vitality observed remained well above the reference value. A control experiment on the effectiveness of H2O2 treatment to initiate stress should be presented. Moreover, the protein CoAlation pattern is diminished after 3 hours of exposure (fig.1B). This point should be investigated or discussed further. Figure 5 shows the results of coalation profiles in the presence or absence of ihnibitor at PRDX2 or 6 and under stress. However, this information should be explained in the legend. In addition, the H2O2 concentrations used should be explained. The role of PRDX2 is clearly demonstrated by the use of conoidin. However, it would be interesting to add a control in the presence of both inhibitors, conoidin and MJ33/ethacrynic acid. Finally, figure 6 shows the CoAlation profile during FCSu-induced capacity. However, this method is not used in fertility clinic and a control by the method used in the clinic would be added (swim up or density gradient) with a sperm motility control and molecular marker of capacitation. 

Author Response

Please, see the attachement with responses to reviewer's comments.

Reviewer 2 Report

The authors show in this manuscript that protein CoAlation is enhanced by oxidative stress in human spermatozoa, with capacitation reducing it. They relate these results to PRDXs activity and motility and viability measurements. This is a well-written manuscript showing novel and relevant results.

There are not especially critical issues, but the authors must clarify statistics (see below, Methods and Results). Some comments on parts are not clear (methods are missing some details).

Even though the original images are provided, the ones in the PDF are of very low quality (blots and plots). Maybe the journal is improving it if accepted? Images and plots should be of excellent quality (300 dpi or better and free of compression artifacts.

L63-65: This could be confusing. Is this thiol oxidation a positive or negative event?

L72: Maybe explain what Tau is? (a microtubule-associated protein in neurons).

Methods:

L363: Revise for minor typos.

L371-2: Detail on buffer and DTT concentration? Centrifugation conditions? It may be explained by 4.7, but this part comes first.

L376: Typo for 10^6.

L378: Concentrations are provided only for some inhibitors.

L383: This isn't very clear. Were samples submitted to hypoosmotic conditions and then analyzed by CASA? Besides clarifying that, please provide details on the buffer and complete details for the CASA system (magnification, frame rate, settings). Were the samples pre-diluted before assessing motility¿

P433: Could you clarify here the factors included in the model? Treatment and time? Were interactions analyzed? Notice that the statistics indicate two-way ANOVA and Tukey test, but, for instance, Table 1 suggests one-way ANOVA and Dunnet's (which have a different purpose). Explain and, if necessary, explain to provide enough information on specific tests.

Results: Clear and well presented.

L83-86: The purpose of reducing vs. non-reducing conditions might be more clearly presented in the introduction (hypothesis).

L94: This part clarifies my observation about L63-65. Maybe this information on the hypothetical purpose of CoAlation could be explained there.

Figures: Indicate if you are showing mean±SD (not a fan of error-bar plots, but adequate). Please confirm that the test is Tukey's (pairwise) or Dunnet's (vs. ref. group, as in Table 1). If you used different post-hoc tests, explain why you changed the data analysis. However, the analysis suggests a one-way ANOVA (a combined time-treatment factor?) and a comparison with the control, not a two-way with a pairwise post-hoc test. A two-way test should produce P values for the two factors involved (not showed) and then post-hoc tests for each factor (and more complex if the interaction is included --it should). Using a single factor with one-way and then a Dunnet's is good, but if a two-way was used, this is not a proper way to show results (and an incomplete one).
Fig. 4: t-BHP legend is lacking in the right-middle panel.

Table 1: This could also be a plot, like figures 1-3. A table has no advantage here, and the change contributes to unnecessary heterogeneity. The footnote says: "Fifth percentile 54% (CI 54-60%)"; what does it mean?

Discussion: Very clear and well done. Only a few comments.

L236: Reversible in protein processing, not in vivo, but I believe the experiment does not test if it happens in vivo. I think the authors mean that their experiment suggests that CoAlation, contrary to other modifications, could be reversible in vivo (to demonstrate fully). This paragraph might need a bit of clarification.

L261: "indicate" -> "help to explain"?
L319: "protein/s"?

Author Response

Please see attachment with responses to reviewer's comments.

Round 2

Reviewer 1 Report

The responses to comments and corrections are correct to accept the manuscript for publication